# Improving the safety of outpatient to Emergency Department transfers: A quality improvement study in a tertiary hospital in Pakistan

Aziza Lakhani[1‡], Samar Fatima[2‡*], Areej Khawaja[2], Qurratulain Virani[3], Muzna Hashmi[2], Tehreem Khan[2], Khairunnissa Hooda[1]

**1** Department of Outpatient Services, Aga Khan University Hospital, Karachi, Pakistan, **2** Department of Medicine, Aga Khan University Hospital, Karachi, Pakistan, **3** Department of Emergency Medicine, Aga Khan University Hospital, Karachi, Pakistan

‡ These authors share first authorship on this work.
* samar.fatima@aku.edu

## Abstract

### Background

Safe and coordinated patient transfers are essential for reducing morbidity, mortality, and adverse events. In outpatient clinics, early recognition of patient deterioration and standardized transfer protocols are key to enhancing safety. This quality improvement initiative addresses these gaps by ensuring the timely identification of critically ill patients, prompt management, and efficient transfer to the emergency department.

### Methods

This study was conducted in two phases. In the pre-implementation phase (August 1–September 14, 2022), a multidisciplinary panel employed a modified Delphi method to revise early warning signs for critically ill clinic patients and developed a structured handoff tool to improve transfer communication. The tool was pilot tested and refined. The implementation phase (September 15–November 30, 2022) included hospital-wide training through webinars and in-person sessions, with effectiveness evaluated in forty staff members using pre- and post-training assessments. The quality initiative, comprising revised early warning criteria and standardized handoff documentation, was formally rolled out on December 1, 2022. Prospective data collection continued for one year (December 2022–November 2023) to evaluate the impact.

### Results

Post-test scores demonstrated significant improvement in staff knowledge, particularly in identifying critically ill patients and initiating appropriate interventions. Among

**Data availability statement:** All relevant data are within the paper and its Supporting information files.

**Funding:** The author(s) received no specific funding for this work.

**Competing interests:** The authors have declared that no competing interests exist.

268 patients requiring transfer, the majority (51.49%) were aged ≥60 years, and 56.3% were male. The most common presenting complaint was acute respiratory distress (31.7%). Compliance with the handoff tool was high (≥70% in 65.6% of cases). However, prolonged emergency department (ED) stays (>7 hours in 45.5% of cases) and a 5.2% mortality rate underscored ongoing challenges in patient flow and triaging.

## Conclusion

Implementing structured transfer protocols, staff training, and standardized handoff tools can significantly improve patient transfer safety and efficiency in outpatient settings. However, further refinements, including enhanced triaging and digitizing documentation practices, are necessary for sustainable improvement. This project highlights the importance of systematic approaches in optimizing intra-hospital transfers in low-resource settings.

## Introduction

Safe and efficient transfers form a crucial juncture in patient care [1]. Patients may require transfer at various stages, either within the same hospital, such as between clinicians or departments (intra-hospital transfers), or between different hospitals (inter-hospital transfers). While the specific reasons for transfer may vary, the overarching goal is consistent: to maintain continuity of care, safeguard the patient during the transition, and enhance overall care quality [2]. Evidence-based clinical practice guidelines outline several key elements for the safe transfer of critically ill patients: the decision to transfer; pre-transfer stabilization and preparation; selection of an appropriate mode of transfer; determination of accompanying personnel; provision of necessary equipment and monitoring during transit; and accurate documentation with a structured handoff at the receiving facility [3]. These principles apply equally to both inter- and intra-hospital transfers.

Without proper organization and careful execution, transfers can pose substantial risks and contribute to adverse outcomes [4,5]. Lauge Sokol-Hessner et al. reported that inter-hospital transfers are associated with approximately 60% longer hospital stays, greater intensive care unit (ICU) utilization, higher costs, lower discharge rates, and increased mortality compared with direct emergency department admissions [6]. Intra-hospital transfers are similarly high risk, with adverse events occurring in up to 70% of cases, life-threatening incidents in 8% and death in 2% of cases [7]. The main contributing factors include communication breakdowns (42% of handoff-related incidents between wards) [8], equipment issues (39%), and staffing deficiencies (61%) [9]. JCIA reports that deficiencies in these processes contribute to 80% of sentinel events [10–12]. These risks can be reduced through the presence of skilled personnel, thorough pre-transfer stabilization, continuous monitoring, and adherence to structured protocols [2].

In high-income countries, the transfer of critically ill patients from outpatient clinics to Emergency Departments (ED) is generally supported by stronger infrastructure, trained personnel, and established protocols, although risks remain [11]. Some settings have shown how targeted quality improvement measures can enhance safety and timeliness. For example, in pediatric cardiac ambulatory care, improved identification and transfer protocols have expedited the movement of critically ill patients [13]. Other studies have demonstrated the role of Rapid Response Teams (RRTs) in promptly managing deteriorating patients in outpatient clinics, thereby facilitating the timely escalation of care and reducing adverse events [14].

However, transfers from outpatient clinics to ED are vulnerable points in the care continuum in the low and middle-income countries (LMICs), where the standardized protocols are often lacking. Intra-hospital transfers, such as these, can lead to undue complications for the patient, including increased risk of morbidity and mortality. As healthcare systems grow increasingly complex and patient transfers become more frequent, strengthening safety and efficiency during these transitions is critical.

To the best of our knowledge, there is a lack of published data from LMICs on the transfer of critically ill patients from outpatient to ED and other high-acuity areas. Building on proven approaches, our quality initiative explores alternative, context-appropriate methods tailored to the resource constraints of an LMIC setting to improve transfer safety and highlight the importance of structured transfer systems in comparable healthcare contexts.

## Objectives

This study aimed to improve the safety, documentation, and tracking of patients transferred from outpatient clinics to the ED and the high acuity areas through a quality improvement approach at Aga Khan University Hospital (AKUH). The specific objectives were:

- To equip clinic staff with the knowledge and skills to detect early warning signs of patient deterioration and take timely action.

- Develop and integrate a formal transfer process flow with clearly defined roles, responsibilities, and equipment readiness checks to ensure safe patient movement.

- Implement a standardized handoff tool for all clinic-to-high-acuity transfers to strengthen communication, ensure accurate documentation, and maintain closed-loop information exchange.

- Track and evaluate transfer activity, including transfer frequency and patient outcomes.

## Methodology

### Context

AKUH is one of Pakistan's leading tertiary care facilities. It is a 710-bed hospital [15] serving a diverse patient population, including many with complex medical conditions, particularly in its outpatient clinics. These outpatient clinics are organized into distinct specialties, including medicine, surgery, oncology, anesthesiology, obstetrics & gynecology, pediatrics, psychiatry, ophthalmology, and family medicine. They are housed in multiple buildings, some of which are two to three stories tall. And most of these buildings are located more than 160 meters away from the ED. These clinics handle a substantial patient volume, with the hospital's outpatient services managing 570,000–600,000 patients annually, including both regular and teleclinic visits.

The process of transferring critically ill patients from outpatient clinics to high-acuity areas such as the ED has been both operationally challenging and clinically unsafe. Over the past two years, a few incidents were reported each quarter, underscoring the persistent risks associated with the transfer of clinically deteriorating patients from outpatient clinics.

Moreover, in the year 2022, a notable increase in such cases was observed, with 180 patients presenting with early warning signs of clinical deterioration opting for care in outpatient clinics rather than the ED. This trend was primarily driven by patient preferences for low-acuity settings, which are perceived as more convenient and cost-effective. However, due to inconsistent documentation and a lack of systematic tracking, precise data on the frequency and nature of these events remain limited.

A root cause analysis revealed major deficiencies in staff knowledge regarding the recognition of and response to patient deterioration, leading to unsafe transfers and increased morbidity. The nursing workforce consisted of registered nurses (RNs) and healthcare assistants (HCAs), with HCAs forming the majority; while skilled in their respective roles, both groups required further training to manage clinical deterioration in outpatient settings. There were no standardized tools, such as the Modified Early Warning Score (MEWS), in place to help identify early signs of patient deterioration. Outpatient clinics lacked standardized protocols and processes to guide the safe transfer and management of critically ill patients. In addition, the absence of a dedicated transfer team, handoff tool, wheelchairs or stretchers, reliable oxygen supply, and ready access to crash carts and lifesaving equipment created significant gaps in emergency preparedness. These limitations left patients vulnerable to delays, fragmented care, and adverse outcomes during transfer to the ED or other high-acuity areas, see Fig 1.

## Quality improvement initiative

The strategies to address these challenges were developed through a structured, iterative approach using established quality improvement (QI) frameworks such as process mapping, stakeholder engagement, and the Plan-Do-Study-Act (PDSA) cycle. Three key solutions were selected based on feasibility, staff capacity, and direct alignment with the identified gaps:

1. Simplified, context-appropriate vital sign parameters (early warning signs) were devised as a new tool, rather than adopting the complex MEWS, to enable rapid identification of deteriorating patients by the nursing staff without delaying care in high-volume outpatient settings.

2. A structured patient transfer process with well-defined roles and responsibilities was implemented to overcome the absence of a dedicated transfer system and team. The on-duty RN was assigned responsibility for ensuring timely patient movement, equipment preparedness, and coordination with the receiving teams.

3. A structured handoff communication tool was implemented to replace unstructured, variable communication during transfers, ensuring that critical clinical details were consistently conveyed to ED staff or high acuity areas to reduce information loss.

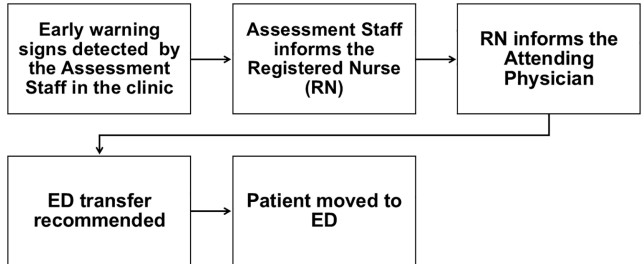

**Fig 1. Illustrates the process of patient transfer from outpatient clinics to the ED before implementation of the quality improvement initiative.**

## Study design and population

This quality improvement project was carried out in two phases. The pre-implementation phase involved development and pilot testing of the tools, while the implementation phase focused on staff training and hospital-wide rollout. Training effectiveness was evaluated using a quasi-experimental, one-group pretest–posttest design in a subset of staff selected through convenience sampling.

In parallel, after the initiative was fully implemented, a prospective observational study was undertaken to evaluate the transfer process in practice and assess associated outcomes. The study population included critically ill adult patients (≥18 years) requiring transfer from outpatient clinics to high-acuity areas, as identified by the treating physician and formally referred to the ED. Outpatient nursing staff responsible for facilitating transfers were also included. Exclusion criteria were transfers from pediatric or obstetric clinics, direct ward admissions, referrals to other healthcare facilities, and patients who declined hospital admission.

## Study phases and timeline

The study was conducted from August 1, 2022, to November 30, 2023, in two phases: pre-implementation and implementation phase.

**Pre-implementation phase: Development and pilot testing of tools (1 August 2022–14 September 2022).** During the pre-implementation phase, a multidisciplinary expert panel comprising consultants from cardiology, pulmonology, internal medicine, and critical care, along with head nurses from outpatient clinics and Rapid Response Team (RRT) specialists, was convened. Using the modified Delphi method, the panel engaged in multiple rounds of structured discussion to develop and refine four key components: early warning signs, the patient transfer process flow, a structured handoff communication tool, and the associated training and assessment materials. This iterative, expert-driven process ensured the content validity of all tools before their feasibility testing and implementation.

The thresholds for abnormal vital signs were revised to reflect the outpatient population, where baseline physiological values often differ from those of inpatients. Unlike acutely unwell hospitalized patients, many patients in the clinic have stable but chronically altered physiology due to multimorbidity or long-term illness. For example, patients with heart failure and reduced ejection fraction may have systolic blood pressures of 80–90 mmHg and a resting tachycardia above 100 beats per minute, while patients with COPD frequently demonstrate higher baseline respiratory rates. The thresholds were therefore tailored to avoid misclassifying such chronically altered parameters as acute deterioration, thereby reducing over-triaging while maintaining sensitivity for detecting meaningful clinical deterioration. The Glasgow Coma Scale (GCS) was retained in its standard form, see Table 1.

In parallel, the panel developed a process flow outlining key steps from early detection to transfer, defining staff roles, communication triggers, and actions to ensure safe and complete handoff. See Fig 2.

To further enhance continuity of care, a structured handoff tool was developed using the SBAR (Situation, Background, Assessment, Recommendation) format [16]. The tool (S1 File) also included sections for the reason of transfer and

**Table 1. Shows the revised vital sign thresholds (early warning signs) to facilitate early detection of clinical deterioration in patients attending the outpatient clinic.**

| Measurable Vital Sign | Newly Set Parameters |
|---|---|
| **Respiratory Rate** (breaths/min) | <10 or>30 |
| **Oxygen Saturation** (%) | <94 |
| **Body Temperature** (°C) | >39 |
| **Blood Pressure** (mmHg) | Hypertension ≥ 170/100<br>Hypotension ≤ 80/60 |
| **Heart Rate** (beats/min) | <60 or>130 |

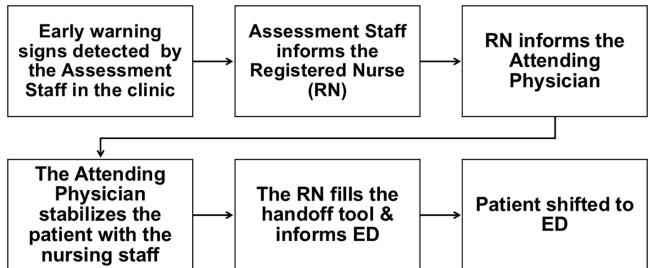

**Fig 2. Illustrates the five preparatory steps required to ensure safe transfer of critically ill patients from outpatient clinics to the emergency department.**

essential interventions performed before transfer, facilitating clear documentation and consistent communication. It was approved by the Health Information Management Services (HIMS) committee and incorporated into confidential patient records.

The quality initiative was piloted in the Medicine outpatient clinic during the first two weeks of September 2022. This high-volume transfer clinic provided an ideal setting to assess the feasibility, reliability, and practical use of the following tools: the revised early warning criteria, the standardized transfer process flow, and the SBAR handoff tool. Staff received structured training and applied the tools during real-time patient transfers. Staff feedback on clarity, usability, and workflow integration was used to refine the tools for consistent use and smooth hospital-wide adoption.

**Implementation phase: Hospital-wide training, full rollout, and evaluation (15 September 2022–30 November 2023).** Following the pilot, hospital-wide training was conducted from 15th September to 30th November 2022. Outpatient nursing staff first attended webinars on early warning recognition and structured handoff, followed by in-person reinforcement sessions led by head nurses for staff nurses and healthcare assistants (HCAs). This staged approach supported consistent adoption across clinics. Training effectiveness was evaluated in a convenience sample of forty staff during one of the sessions. Participants completed a pre-training questionnaire (10 items on early warning signs, patient management, and handoff communication), attended training on the revised warning signs, transfer process flow, and the newly developed handoff tool, and then repeated the same questionnaire immediately after training to assess knowledge gains.

On 1st December 2022, the quality initiative was formally implemented across the hospital, and prospective data collection on all patient transfers began, continuing through November 2023. Clinics formally adopted the revised early warning criteria and standardized handoff documentation as routine practice.

To strengthen preparedness, equipment safety checks were instituted, and emergency transfer kits stocked with essential medications and supplies were distributed to clinics (see Table 2). Both phases are depicted in Fig 3.

**Sustainability measures and ongoing staff competency reinforcement.** To ensure the sustainability of the quality initiative, the training program is incorporated into the orientation of newly recruited nurses. For existing staff, quarterly refresher sessions are scheduled to reinforce critical competencies in identifying early warning signs, using the standardized handoff tool, and adhering to the established patient transfer protocol. In the future, we also plan to digitize the handoff process to further enhance efficiency and continuity of care.

## Measures

The primary outcome of this quality improvement initiative was patient safety, assessed through adverse events during outpatient-to-ED transfers and in-hospital mortality. Adverse events were defined as any unintended clinical incident occurring during the transfer process, including clinical deterioration, delays in care, or errors in documentation or

**Table 2. Details the equipment safety checks and standardized contents of the emergency transfer kit, including readiness, functionality, and essential supplies required during the transfer of critically ill patients.**

| Equipment safety checks | Standardized contents of the emergency transfer kit |
|---|---|
| Stretcher | Injection atropine, epinephrine, and midazolam |
| Cardiac monitor | 0.9% normal saline (25 ml) |
| Emergency transfer kit | Syringes (3 ml and 5 ml) |
| Filled Oxygen Cylinder (if required) | IV cannula (20G, 22G, 24G) |
| Intravenous Access and Intravenous Hydration (if required) | Tegaderm, long Line, alcohol swabs |
| Defibrillator (if required) | Paper tape, tourniquet |
| Transfer checklist | A pair of gloves |

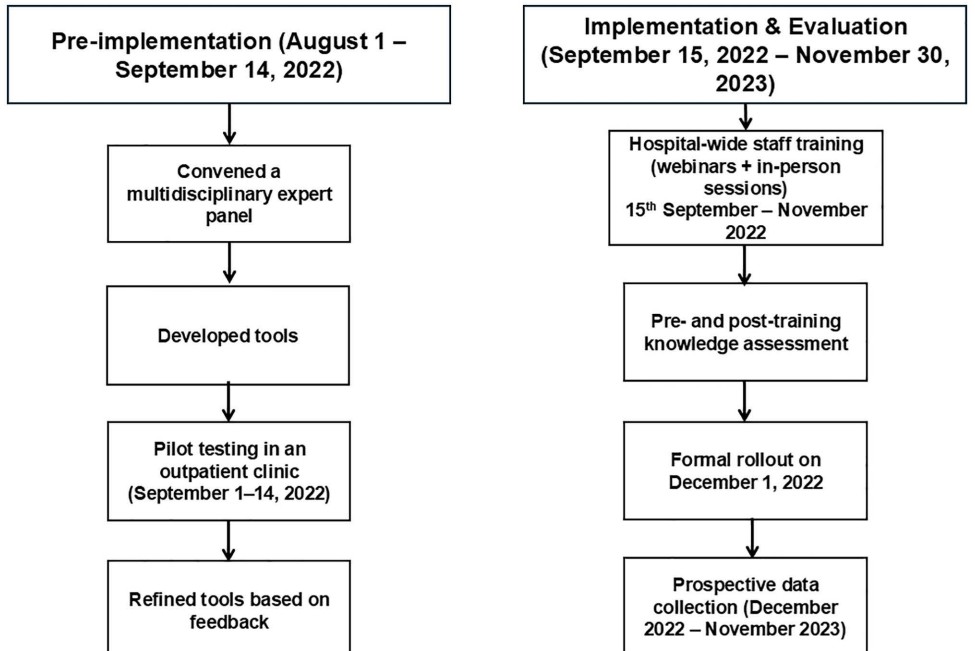

**Fig 3. Illustrates both the pre-implementation and implementation phases of the quality improvement initiative for patient transfer from outpatient clinics to the emergency department.**

communication. These events were captured through the institutional incident reporting system and notifications from the receiving team. In-hospital mortality was defined as death occurring during the same hospital admission as the ED visit, with data extracted from electronic health records. Secondary outcomes included ED length of stay (LOS) and patient disposition. ED LOS was defined as the time from patient transfer in the ED to inpatient admission or discharge from the ED, and patient disposition was categorized as ward admission, ICU admission, or discharge directly from the ED, based on transfer records.

Process measures evaluated the fidelity of the quality initiative, specifically compliance with the standardized SBAR handoff tool during transfers. Compliance was defined as the proportion of transfers in which all SBAR components were documented and was monitored monthly through structured audits by trained staff using standardized checklists.

Balancing measures tracked potential unintended consequences, such as workflow delays or additional staff burden during outpatient-to-ED transfers, identified through discussion in Comprehensive Unit-based Safety Program (CUSP) meetings and the institutional incident reporting system.

## Ethical considerations

This project utilized two data sources: (1) pre- and post-test assessment data from staff who participated in the training, for which informed written consent was obtained, and (2) de-identified patient data and outcomes extracted from the handoff communication tool, patient files, and electronic medical records. Confidentiality and anonymity were strictly maintained, and no patient-identifiable information was recorded. No live interviews or direct patient interactions were conducted.

Permission to conduct this project was obtained from the Chief Medical Officer (CMO) office and the Ethical Review Committee (ERC) of AKUH, Karachi, Pakistan, on 27 August 2023 (IRB reference number: 2023-8537-26185). The authors confirm that this manuscript is original and has not been published or submitted elsewhere.

## Statistical analysis

The statistical analyses were conducted using Stata version 16. Descriptive statistics, including means, standard deviations, frequencies, and percentages, were used to characterize the dataset. To assess whether the missing data was missing completely at random (MCAR), Little's MCAR test was performed. The test results ($\chi^2 = 4.8987$, $p = 0.2979$) indicated that the pattern of missingness was random and did not introduce systematic bias. Therefore, no imputation was performed.

To evaluate differences between pre- and post-test scores of participants, paired t-tests were utilized. These tests assessed the effectiveness of the training program by comparing pre- and post-test results to understand participants' comprehension of the handoff communication tool. In addition to assessing the practical significance of pre- and post-test differences, effect sizes were calculated using Cohen's d, a widely used measure for standardized mean differences. According to Cohen's guidelines, effect sizes are categorized as follows: small ($d = 0.2$), moderate ($d = 0.5$), and large ($d = 0.8$). These thresholds provide a meaningful interpretation of the magnitude of change beyond statistical significance.

Before performing the paired t-tests, the Shapiro-Wilk test was conducted to check for normality, and the normality assumption was met. We also reported 95% confidence intervals for the mean differences. We did not apply the Bonferroni correction for multiple comparisons to allow for the identification of potential trends.

Following the large-scale implementation of the program, we analyzed patient transfer data to evaluate the frequency and outcomes of patient transfers. Descriptive statistics for patient demographics and characteristics, presenting complaints requiring emergency transfer, and comparisons of vitals before and after transfer were calculated.

## Results

### Pre- and post-test assessment

A total of 40 nursing staff participated in the workshop; most were female (77.5%, n = 31), while males accounted for 22.5% (n = 9). Participants had a mean professional experience of 12.3 ± 7.85 years. More than half were registered nurses (55%, n = 22), while the remainder were nurse instructors (22.5%, n = 9) or had other designations (22.5%, n = 9).

Following the training, participants exhibited significant improvements in eight out of ten knowledge and decision-making questions. The most substantial gains were observed in the assessment of a drowsy patient (p = 0.005, d = 0.47) and the criteria for transfer from the outpatient department to the emergency department (p = 0.018, d = 0.39). Other areas showing significant improvement included the required transfer equipment, description of the transfer process, management of a drowsy patient in the waiting area, response to a drowsy patient with hypotension, and identification of patient findings necessitating

ED transfer (all p<0.05, d=0.17–0.39). Two items did not demonstrate statistically significant changes, likely due to higher baseline knowledge in those areas (see Table 3).

A box plot of the pre- and post-test results showed that participants' scores improved following the training program. The median post-test score was higher than the pre-test median, indicating an overall improvement in knowledge. The pre-test scores displayed a wider spread, with lower whiskers extending further down, reflecting a broader range of initial knowledge levels. In contrast, the post-test distribution was more concentrated, demonstrating that the training effectively standardized participants' ability to recognize early warning signs and follow appropriate patient transfer protocols (see Fig 4).

## Outcomes of patient transfers following the implementation of the quality improvement initiative

Between December 1, 2022, and November 30, 2023, approximately 268 patients required transfer to the ED and other high-acuity areas. Of these, 51.5% (n=138) were aged 60 years or older, and the majority were male (56.3%, n=151). On presentation, 39.18% (n=105) of patients had a systolic blood pressure between 100–135 mmHg, while 22.7% (n=61) had a systolic pressure ≤99 mmHg. In 23.8% (n=64) of cases, the diastolic blood pressure was ≤58 mmHg. Heart rates were within 83–119 beats per minute (45.15%, n=121), while respiratory rates (RR) were ≤18 breaths per minute in 40.67% (n=109) of patients, and >22 breaths per minute in 36.57% (n=98). Most patients were afebrile (84.3%, n=226), although 10.45% (n=28) presented with high-grade fever (>39°C). Oxygen saturation was within the normal range (≥95%) in 67.54% (n=181) of patients, whereas 32.46% (n=87) had levels <94%. The most common comorbidities among those transferred were malignancy (24.25%, n=65) and a combination of diabetes and hypertension (19.4%, n=52). Among those who required transfer, 48.88% (n=131) were registered as follow-up visits, while 44.03% (n=118) were initial clinic visits. Most patients were transported either by stretcher (48.13%, n=129) or on foot (34.7%, n=93). And the most common presenting complaint requiring transfer was acute respiratory distress or compromised airway (31.72%, n=85), followed by acute abdominal pain, vomiting, or diarrhea (22.01%, n=59), see Table 4.

**Table 3. Shows the pre-test and post-test scores on knowledge and decision-making questions among nursing staff.**

| Questions | Pre-Test Mean | Post-Test Mean | Mean Difference (SD) | Cohen's d Effect Size | 95% Confidence Interval | P-Value |
|---|---|---|---|---|---|---|
| A 48-year-old female called for assessment. The nurse assessed that the patient was drowsy. What will be your first intervention? | 0.65 | 0.88 | −0.23 (0.48) | −0.47 | [-0.378, -0.072] | 0.005 |
| Under what circumstances would a patient require transfer from the Outpatient Department to the ED? | 0.68 | 0.9 | −0.23 (0.58) | −0.39 | −0.409, −0.041] | 0.018 |
| Mark all the equipment required for transferring the patient from the outpatient to the ED. | 0.6 | 0.8 | −0.20 (0.52) | −0.39 | [-0.365, -0.035] | 0.019 |
| Please describe the transfer process from the Outpatient department to the ED. | 0.25 | 0.48 | −0.23 (0.62) | −0.36 | [-0.423, -0.027] | 0.027 |
| In the clinic waiting area, you observe a young patient sleeping for at least 1 hour. What will be your intervention? | 0.83 | 0.98 | −0.15 (0.43) | −0.35 | [-0.286, -0.014] | 0.032 |
| A 45-year-old female with a blood pressure of 80/40 mmHg needs transfer to the ER. What will be your first step? | 0.6 | 0.78 | −0.18 (0.55) | −0.325 | [-0.351, 0.001] | 0.051 |
| A patient is drowsy but arousable with a BP of 90/60 mmHg. What would be the appropriate next step? | 0.63 | 0.9 | −0.28 (0.55) | −0.27 | [-0.452, -0.098] | 0.003 |
| A 65-year-old male with active rectal bleeding is to be referred to gastroenterology. What will you do? | 0.45 | 0.6 | −0.15 (0.62) | −0.24 | [-0.349, 0.049] | 0.135 |
| A 35-year-old male fainted and is unresponsive during consultation. What will be your first intervention? | 0.98 | 0.9 | 0.08 (0.35) | −0.21 | [-0.037, 0.187] | 0.183 |
| Which patient findings require transfer to the ED? | 0.35 | 0.53 | −0.18 (0.45) | −0.17 | [-0.318, -0.032] | 0.018 |

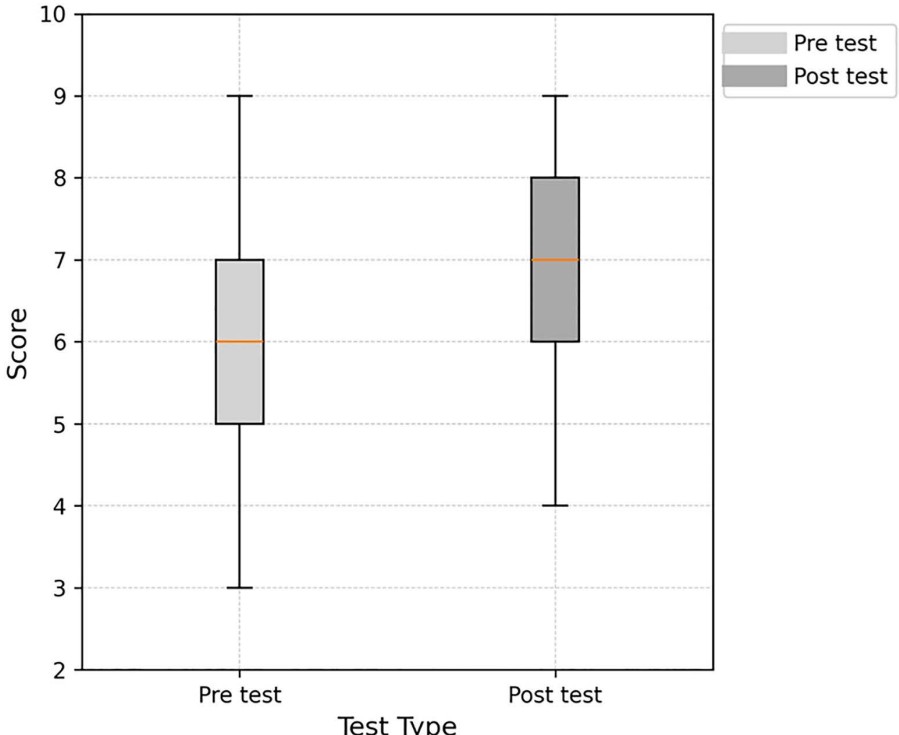

**Fig 4. Illustrates staff pre-test and post-test scores following training using a box plot.**

The comparison of vital signs before and after patient transfer revealed statistically significant improvements in heart rate and respiratory rate. The mean heart rate decreased by 1.87 (SD = 14.57, p = 0.04), and the mean respiratory rate decreased by 1.008 (SD = 5.119, p < 0.001). However, changes in other parameters were not statistically significant, see Table 5.

The majority of patients (89.55%, n = 240) were transferred to the ED. Regarding patient disposition from the ED, 30.22% (n = 81) were transferred to the Special Care Unit (SCU) and 29.48% (n = 79) to the general ward. Additionally, 15.63% (n = 41) of the patients left against medical advice (LAMA), and 13.43% (n = 36) were discharged directly from the ED. The length of stay in ED varied, with the majority (45.52%, n = 122) staying for more than 7 hours. Emergency procedures were performed in 24.63% (n = 66) of cases. A total of 5.22% (n = 14) of patients died during their stay in the hospital. The handoff tool was completed for more than 70% of transfers in 65.67% (n = 176) of the cases. During the study period, no significant workflow delays or additional staff burden were reported, based on CUSP discussions, incident reports, or emails from the staff or the receiving teams (see Table 6).

Transfer patterns showed an initial rise following staff training, possibly due to increased vigilance as newly trained personnel became more proactive in identifying and escalating patients, followed by a period of stable, lower transfer levels, which may indicate that the process had settled into a new norm with more accurate triage and improved care delivery within the outpatient setting, see Fig 5.

## Discussion

Timely recognition, stabilization, and safe transfer of deteriorating patients are critical for patient safety. Our quality improvement initiative streamlined transfers from outpatient to emergency care using a "stabilize and shift" approach with

**Table 4. Shows the demographic characteristics and presenting complaints of patients (n = 268) requiring transfer from the outpatient clinic to the ED or other high-acuity areas.**

**Baseline characteristics of patients requiring transfer from the outpatient clinic to the high acuity area**

| Variable | Frequency n (%) |
|---|---|
| **Age** | |
| ≥18 & <39 | 69 (25.75) |
| 39 & <60 | 61 (22.76) |
| ≥60 | 138 (51.49) |
| **Gender** | |
| Male | 151 (56.34) |
| Female | 117 (43.66) |
| **BP systolic on presentation** | |
| ≤99 | 61 (22.76) |
| >99 & <135 | 105 (39.18) |
| >135 | 102 (38.06) |
| **BP diastolic on presentation** | |
| ≤58 | 64 (23.88) |
| >58 & ≤79 | 120 (44.78) |
| >79 | 84 (31.34) |
| **Heart rate** | |
| ≤82 | 67 (25) |
| >82 & <120 | 121 (45.15) |
| ≥120 | 80 (29.85) |
| **Respiratory rate** | |
| ≤18 | 109 (40.67) |
| >18 & <22 | 61 (22.76) |
| >22 | 98 (36.57) |
| **Temperature** | |
| <38 | 226 (84.33) |
| 38-39 | 14 (5.22) |
| >39 | 28 (10.45) |
| **SpO2** | |
| Abnormal <95 | 87 (32.46) |
| Normal 95–100 | 181 (67.54) |
| **Comorbidity** | |
| Malignancy | 65 (24.25) |
| DM + HTN | 52 (19.4) |
| Other comorbidities | 43 (16.04) |
| Cardiovascular diseases | 30 (11.19) |
| DM + HTN + CKD | 29 (10.82) |
| Surgical conditions | 19 (7.09) |
| COPD | 16 (5.97) |
| Endocrine conditions | 11 (4.1) |
| No comorbidities | 3 (1.12) |
| **Type of visit** | |
| Follow-up | 131 (48.88) |

*(Continued)*

**Table 4.** (Continued)

**Baseline characteristics of patients requiring transfer from the outpatient clinic to the high acuity area**

| Variable | Frequency n (%) |
|---|---|
| Initial | 118 (44.03) |
| Walk-in | 14 (5.22) |
| Missing data | 5 (1.87) |
| **Mode of transportation to the ED** | |
| Stretcher | 129 (48.13) |
| On foot | 93 (34.7) |
| Wheelchair | 46 (17.16) |

**Presenting complaints of the patients requiring transfer from the outpatient clinic to the high acuity area**

| Presenting Complaint | Frequency n (%) |
|---|---|
| Acute respiratory distress or compromised airway | 85 (31.71) |
| Acute abdominal pain, vomiting, or diarrhea | 59 (22.01) |
| Hemodynamic instability with altered vital signs | 26 (9.70) |
| Acute neurological deterioration or change in mental status from baseline | 25 (9.30) |
| Others | 22 (8.20) |
| Acute chest pain with suspicion of cardiac arrest | 20 (7.46) |
| Any abnormal and critical test that requires urgent management and further monitoring | 18 (6.71) |
| Significant active bleeding from any source | 13 (4.85) |

**Table 5.** Shows the vital signs before and after patient transfer, including mean values and statistical significance.

| Variable | Mean difference | Standard deviation | P value |
|---|---|---|---|
| **Heart rate** | −1.87 | 14.57 | 0.04 |
| **Respiratory rate** | −1.008 | 5.119 | 0.00 |
| **Temperature** | 0.186 | 4.35 | 0.51 |
| **SpO2** | 0.54 | 8.61 | 0.32 |
| **Systolic blood pressure** | −0.89 | 17.87 | 0.41 |
| **Diastolic blood pressure** | −1.008 | 12.60 | 0.21 |

a structured five-step transfer process, supported by logistical adjustments, collaboration with resuscitation and administrative teams, staff training on revised early warning signs, and standardized handoff tools. These interventions improved the recognition and timely transfer of acutely unwell patients, most of whom were older adults with acute respiratory distress and multiple comorbidities, highlighting the vulnerability of this group and the need for targeted acute care strategies in outpatient settings. Despite these improvements, prolonged ED stays, and some adverse outcomes persisted, reflecting ongoing system-level challenges in patient flow and acute care delivery.

Pakistan's healthcare system, like those of many low- and middle-income countries, differs markedly from systems in developed nations such as the UK and the USA. In high-income settings, healthcare is typically structured around primary care, with general practitioners (GPs) serving as the first point of contact and coordinating referrals to specialists and tertiary services. By contrast, primary care in Pakistan is less organized, leading many patients to bypass GPs and

**Table 6. Details the outcomes of patients transferred from the outpatient clinic to the ED or other high-acuity area.**

| Variable | Frequency n (%) |
|---|---|
| **Area to be transferred** | |
| ED | 240 (89.55) |
| SCU | 19 (7.09) |
| CCU, ICU, and Others | 9 (3.36) |
| **Patient disposition from the ED** | |
| SCU | 81 (30.22) |
| Ward | 79 (29.48) |
| CCU | 21 (7.84) |
| ICU | 3 (1.12) |
| LAMA | 41 (15.63) |
| Discharged | 36 (13.43) |
| Missing Data | 7 (2.28) |
| **Length of ED stay (hours)** | |
| <=5 | 51 (19.03) |
| >5 and <7 | 57 (21.27) |
| >7 | 122 (45.52) |
| Not Applicable | 38 (14.18) |
| **Emergency procedure required** | |
| No | 202 (75.37) |
| Yes | 66 (24.63) |
| **Patient expiry** | |
| Yes | 14 (5.22) |
| No | 254 (94.78) |
| **Handoff tool filled out (%)** | |
| >70 | 176 (65.67) |
| 50-70 | 74 (27.61) |
| <50 | 18 (6.72) |

seek specialist care directly in tertiary care facilities [17]. Limited health literacy and financial constraints further shape care-seeking behaviors, as outpatient consultations are often perceived as more affordable and accessible than emergency department visits [18–20]. This may partly explain our finding that many critically ill patients first presented to outpatient clinics rather than emergency departments, often after traveling long distances from rural areas, and subsequently deteriorated while in the clinic [21]. In such contexts, evidence suggests that structured transfer protocols, trained staff, and clear communication pathways are essential to minimize risks and ensure timely stabilization.

However, there is limited published evidence specifically addressing the transfer of patients from outpatient clinics to emergency departments and other high-acuity areas. Only a few studies have explored this area, suggesting the need for more research to optimize safe transfer practices in such settings. One example is a pediatric quality improvement initiative that enhanced the safety and efficiency of ambulatory cardiology admissions. In this project, outpatient monitoring was strengthened through the Children's Hospital Early Warning Score (CHEWS), which enabled nurses to systematically assess patient status, detect early signs of deterioration, and activate escalation protocols when necessary. To support structured communication, handoff was facilitated through a "virtual huddle," guided by the I-PASS framework (Illness severity, Patient summary, Action list, Situation awareness and contingency planning, and Synthesis by receiver). This

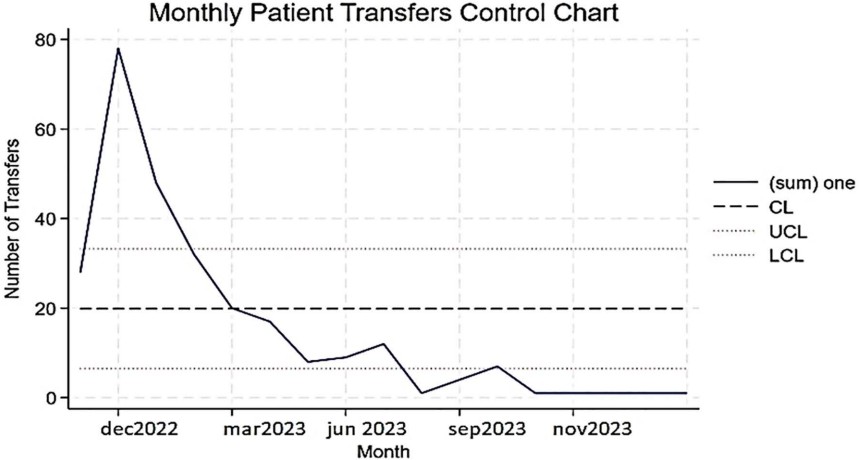

**Fig 5. Shows a control chart of monthly patient transfers.** The solid line indicates the observed number of transfers, the dashed line represents the center line (mean), and the dotted lines indicate the upper and lower control limits.

standardized approach reduced miscommunication and allowed for the timely prioritization of "acuity patients" over stable patients awaiting monitored beds. Together, these interventions led to measurable improvements in patient safety and process efficiency [13]. However, in our study, the MEWS was excluded, as explained in the rationale, and the adoption of tools such as I-PASS was not feasible in the outpatient clinic due to staffing constraints, limited clinical training, and high patient volumes that restricted time for additional and detailed documentation. Instead, we implemented SBAR [22], which was already familiar to nursing staff from its use in the inpatient setting (see S1 File). The form captures essential information, including presenting complaints, comorbidities, vital signs, infection status, and plan of care, aligning with the WHO Safer Handover Toolkit [23]. While the WHO toolkit is primarily intended for inpatient settings, we adopted these recommendations for outpatient-to-ED transfers by including additional elements such as transportation mode, patient accompaniment, and infection precautions. Systematic review evidence suggests that SBAR can improve the clarity and completeness of handoff communication and enhance patient safety outcomes. However, the review also noted that the impact on patient outcomes was inconsistent across studies, and its effectiveness depends on context, staff training, and adherence [16].

Other studies focusing on outpatient-to-ED transfers have shown the effectiveness of Rapid Response Team (RRT) systems [14,24], where advanced practice providers, nurses, and administrators stabilize patients and ensure safe transfers. Early warning criteria such as vital signs, altered mental status, and respiratory distress have been used to trigger escalation. In contrast, our expert panel assessed the feasibility of introducing an RRT model in our outpatient clinic but could not reach consensus due to human resource limitations. At our institution, the Rapid Response Team (RRT) consists of a nurse, an ICU fellow or senior medical officer (SMO), and a consultant who provides hospital-wide critical care. Nursing staff rotate coverage each shift. In the mornings, the team is staffed by an ICU fellow or SMO alongside a nurse. Evenings and nights are managed by on-call ICU staff with nurse support, while the consultant is available on-site until the evening and by phone thereafter. Given these staffing limitations and the team's existing workload, extending RRT coverage to the outpatient clinic was considered impractical.

Beyond team logistics, our study also highlights key patient characteristics necessitating transfer. As stated earlier, geriatric patients with multimorbidity represent a vulnerable group requiring prioritized and timely interventions to prevent deterioration. These findings align with existing literature that identifies older adults with chronic conditions as a high-risk group requiring prompt acute care interventions [25]. This underscores the critical need to minimize waiting times for

geriatric patients and those with multimorbidity to prevent clinical deterioration and adverse outcomes. Although most vital signs were within normal limits upon presentation, observed abnormalities in respiratory rates, likely associated with acute respiratory distress, a common presenting symptom, underscore the need for heightened vigilance in respiratory monitoring.

A noteworthy finding was the high proportion of patients requiring transfer to the ED, with fewer being directly transferred to the SCU, CCU, or ICU. While the ED often serves as the first point of care for acute cases, managing critically ill patients there is resource-intensive [26,27]. Additionally, half of the patients spent more than seven hours in the ED before transferring to a specialized unit, reflecting delays in accessing definitive care. Prolonged ED stays are clinically significant, as hospital inpatient admissions exceeding five hours from ED arrival have been associated with increased all-cause 30-day mortality [28]. To address these delays, efforts should focus on facilitating direct transfers to specialized units whenever appropriate.

Although overall compliance with the handoff tool was relatively high, documentation inconsistencies remained a concern [29]. At AKUH, this project has been successfully integrated into routine practice, supported by ongoing monitoring and regular staff training to maintain the standards achieved. Evidence from other settings further highlights the value of structured training: for example, a quasi-experimental study in Zahedan, Iran, demonstrated that training nurses in safe transfer using a checklist significantly improved the quality of intrahospital handoffs, with intervention group scores markedly higher than controls [30]. For future steps, digitization of handoffs is recommended to enhance patient safety and continuity of care. While physicians already use electronic handoffs, nurses still rely on handwritten SBAR notes. Full EHR implementation, planned within the next one to two years, will allow complete digitization and long-term sustainability. As an interim step, integration of nursing assessments and early warning tools into the "*My Patient*" application can provide real-time alerts and systematic documentation. Regular sustainability audits are essential to ensure compliance, close gaps, and drive continuous quality improvement. Together, these measures could standardize handoffs across departments and offer a scalable model for resource-limited healthcare settings.

Nkhwashu *et al.* (2023) identified several facilitators and barriers in implementing quality improvement programs (QIPs). Key facilitators included staff commitment, management support, capacity building, multidisciplinary collaboration, and external partnerships. Conversely, barriers included a lack of ownership, resistance to change, low motivation, unclear strategies, weak leadership, and limited resources [31]. Although QI initiatives often encounter challenges such as staff resistance, difficulty securing buy-in, or form fatigue, our project did not experience these issues. This can be attributed to the initiative's integration within the Comprehensive Unit-based Safety Program (CUSP) team of the outpatient department, which fostered clear accountability, strong leadership, dedicated champions, and organizational support—factors identified by the Consolidated Framework for Sustainability in Healthcare as critical for long-term success [32,33]. Through CUSP's structured engagement, 2–3 staff from each clinic area actively participated, leadership provided visible support, and monthly meetings ensured regular review and reinforcement. Consequently, staff adopted the revised early warning signs and structured handoff tools, embedding the transfer protocol into routine practice. By aligning with these sustainability principles, our approach not only addressed barriers reported elsewhere but also demonstrated a low-cost, scalable, and sustainable model for safe transfers in resource-limited settings.

Overall, this quality initiative was cost-effective, with minimal additional expenses primarily related to staff training and procurement of equipment such as movable examination couches. The costs were managed within the outpatient department's existing budget, eliminating the need for external funding. Additionally, the project was associated with an increased recognition of deteriorating patients, which may have contributed to the improved outcomes. This effort introduced a structured approach to improving patient transfers in a low-acuity outpatient setting through enhanced communication tools, standardized patient identification, and processes modified to meet the specific needs of outpatient-to-emergency transitions, along with the initiation of systematic tracking and monitoring of such cases.

## Limitations

Although this quality initiative was associated with positive outcomes, several limitations must be considered. The study was conducted at a single institution with a small sample size, which may limit the generalizability of findings to other healthcare settings with different patient populations or resources. The absence of a control group and the lack of prior data for comparison prevent definitive conclusions about causality, and external factors may have influenced the observed improvements. While the tools were tested for feasibility and consistency during the pilot phase, no formal statistical reliability testing was performed, and the subjective nature of data interpretation may have introduced bias, particularly in assessing adherence to handoff protocols and the effectiveness of the initiative. Variability in the implementation of safety measures across teams or shifts may have further contributed to inconsistencies in outcomes. Additionally, external factors such as fluctuations in patient volume, staffing shortages, bed availability, and unexpected departmental disruptions were not systematically analyzed but could have significantly influenced transfer dynamics. The observed improvements in the knowledge suggest a positive impact of the training; however, the effect sizes, which range from small to moderate, may reflect differences in baseline knowledge among participants as well as limitations of the assessment tool. The tool was constrained by the small number of items and variable difficulty, with some questions showing ceiling effects that reduced the ability to detect change. Despite the modest magnitude, these improvements may meaningfully enhance recognition of early warning signs and adherence to patient transfer protocols. Future studies should address these limitations by incorporating multi-institutional designs, prolonged follow-up periods, and objective assessment methods, such as independent audits or automated tracking systems, to evaluate both the durability of improvements and the broader applicability of findings across diverse healthcare settings.

## Conclusion

This quality improvement initiative demonstrates the practicality and benefits of structured transfer protocols and targeted staff training in enhancing patient safety and transfer efficiency in outpatient settings, particularly in low-resource contexts. Structured communication, physical accompaniment, and stakeholder collaboration significantly improved the safety of patient transfers from outpatient clinics to the ED. This approach is low-cost, sustainable, and scalable to similar settings. Further research, including controlled studies and broader evaluations, is needed to validate these findings and identify strategies for sustaining and expanding these improvements.

## Supporting information

**S1 File. Handoff communication tool for patient transfers from outpatient clinics to high-acuity care areas.**
(DOCX)

**S1 Data. Dataset containing individual patient data on the outcomes of transferred patient included in the study.**
(XLS)

**S2 Data. Dataset containing pre- and post-test assessments of nursing staff who participated in the training session on early warning recognition and structured handoff.**
(XLS)

## Acknowledgments

The authors extend their sincere gratitude to the Director of Outpatient Services for facilitating the implementation process by providing logistical support, including the arrangement of movable gurneys for the clinics. Appreciation is also extended to the expert panel and the Rapid Response Team (RRT) for their valuable input on the revised vital sign parameters and the handoff tool. The contribution of front-line nursing staff and nursing leads in the outpatient clinics during data collection is gratefully acknowledged.

## Author contributions

**Conceptualization:** Aziza Lakhani, Samar Fatima.

**Data curation:** Aziza Lakhani, Samar Fatima, Qurratulain Virani, Tehreem Khan.

**Formal analysis:** Aziza Lakhani, Samar Fatima, Muzna Hashmi.

**Investigation:** Aziza Lakhani, Samar Fatima, Areej Khawaja, Qurratulain Virani, Muzna Hashmi, Tehreem Khan.

**Methodology:** Aziza Lakhani, Samar Fatima, Qurratulain Virani, Muzna Hashmi.

**Project administration:** Aziza Lakhani, Samar Fatima, Qurratulain Virani, Tehreem Khan.

**Software:** Muzna Hashmi.

**Supervision:** Aziza Lakhani, Samar Fatima, Khairunnissa Hooda.

**Validation:** Aziza Lakhani, Samar Fatima, Muzna Hashmi, Khairunnissa Hooda.

**Visualization:** Aziza Lakhani, Samar Fatima, Areej Khawaja, Qurratulain Virani, Muzna Hashmi, Tehreem Khan, Khairunnissa Hooda.

**Writing – Original Draft Preparation:** Samar Fatima, Areej Khawaja, Qurratulain Virani.

**Writing – review & editing:** Samar Fatima, Areej Khawaja, Qurratulain Virani, Muzna Hashmi.

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
