## [Decision Letter · Decision Letter 0]

30 Apr 2025

Dear Dr. Fatima,

Thank you for submitting your manuscript to PLOS ONE. After careful consideration, we feel that it has merit but does not fully meet PLOS ONE’s publication criteria as it currently stands. Therefore, we invite you to submit a revised version of the manuscript that addresses the points raised during the review process.

**ACADEMIC EDITOR:**

I have reviewed the manuscript and consider it suitable for publication, pending the resolution of several minor revisions.

Minor concerns:

Add a short paragraph in the Methods section noting the lack of control group, which is only mentioned later in the Limitations.Clarify whether the handoff tool was validated (face/content validity, pilot feedback, etc.) before implementation.While p-values are provided, effect sizes for pre-post test differences should be added to convey practical significance.Explicitly state how missing data (e.g., 2.28% in Table 7) were handled in analysis.Figures (1–4) are referenced but were not included in the PDF for evaluation (they are in separate TIFF files). Ensure that these are reviewed and captions are descriptive enough to be interpreted independently.The handoff tool (S1 File) should be briefly described (e.g., sections, items) in the text when first introduced.Minor polishing needed for consistent tenses and clarity. For example: "Patients were predominantly transported by stretcher (48.13%) or walking (34.7%)" → "on foot" might be clearer. “...hospital-wide over the next two to three months” → specify actual timeline if possible.Strengthen discussion with comparative insights from other QI projects in LMICs or outpatient-to-ED transition studies, if available.Clarify whether the training program has ongoing refresher modules or sustainability plans post-study.

We look forward to receiving your revised manuscript.

Kind regards,

Mohd Ismail Ibrahim, MCom.Med

Academic Editor

PLOS ONE

Journal Requirements:

Reviewers' comments:

Reviewer's Responses to Questions

**Comments to the Author**

1. Is the manuscript technically sound, and do the data support the conclusions?

Reviewer #1: Partly

2. Has the statistical analysis been performed appropriately and rigorously?

Reviewer #1: No

3. Have the authors made all data underlying the findings in their manuscript fully available?

Reviewer #1: Yes

4. Is the manuscript presented in an intelligible fashion and written in standard English?

Reviewer #1: Yes

Reviewer #1: Dear Author(s),

Title Improvement Needed: The title of the manuscript could be enhanced for clarity and relevance.

Abstract

Method Clarity: The methodology is not clearly articulated in the abstract. A more precise definition of the phases and techniques used is necessary.

Introduction

Problem Justification: The significance of the research problem should be elaborated upon. A clearer explanation of why this study is necessary would strengthen the introduction.

Methodology

General Explanation: A general overview of the phases should be presented in a dedicated figure within this section.

Method Explanation: The overall methodology lacks clarity. A more thorough explanation is needed to understand the study's design and execution.

Discussion

Findings-Based Discussion: The discussion section should initiate with a summary of the primary findings from the study.

Comparative Analysis: Include comparisons with countries having similar income levels or employing analogous methods, such as India and Iran. This context would enrich the discussion and provide a broader perspective.

Presentation

Schematic Representation: Processes should be illustrated in a schematic form to aid understanding.

Conclusion

Ethical Declaration: A concluding declaration should be added to address ethics, budget considerations, author contributions, acknowledgments, and any necessary thanks.

**Do you want your identity to be public for this peer review?** For information about this choice, including consent withdrawal, please see our Privacy Policy

Reviewer #1: No

---

## [Author Response · Author response to Decision Letter 1]

9 Jun 2025

Response to Reviewer

Subject: Revised Manuscript Submission – PONE-D-25-12396

Dear Dr. Mohd Ismail Ibrahim,

Thank you for considering our manuscript (ID: PONE-D-25-12396), entitled “Rapid response and safety optimization: enhancing critical transfers from outpatient clinics to high acuity areas – a quality improvement project” for review.

We are grateful for your and the reviewers’ insightful comments. In response, we have carefully revised and modified our manuscript in accordance with the reviewers' suggestions. We believe these revisions have strengthened the manuscript and hope that it is now suitable for publication in PLOS ONE.

In accordance with the journal's guidelines, we are submitting the following files:

• A marked-up copy of the revised manuscript with tracked changes, labeled "Revised Manuscript with Track Changes".

• A clean, unmarked version of the revised manuscript, labeled "Manuscript".

We appreciate your time and consideration, and we remain available to address any further questions or concerns you may have.

Yours sincerely,

On behalf of all authors,

Dr. Samar Fatima

ACADEMIC EDITOR COMMENTS:

Comment 1:

Add a short paragraph in the Methods section noting the lack of control group, which is only mentioned later in the Limitations.

Reply:

As per your suggestion, the following paragraph has been added to the Methods section under the heading Study Design, Setting, and Data Source, addressing the absence of a control group.

For ease of reference, the paragraph is provided below:

" A control group was not included in the study due to the operational and ethical challenges of withholding a potentially beneficial intervention from a subset of patients and staff. Given the quality and safety implications associated with patient transfers, it was deemed necessary to implement the intervention across the entire relevant cohort to ensure standardized practice and minimize risk."

Comment 2:

Clarify whether the handoff tool was validated (face/content validity, pilot feedback, etc.) before implementation

Reply:

The handoff tool was validated prior to implementation using a modified Delphi method to establish content validity. It was piloted during Phase One of the study. Feedback from this pilot led to minor revisions to enhance clarity and usability before final implementation. The Phase One section in the manuscript has also been revised for improved clarity.

Comment 3:

While p-values are provided, effect sizes for pre-post test differences should be added to convey practical significance.

Reply:

We appreciate the reviewer's insightful comment. In response, we have incorporated Cohen's d values in Table 5 to complement the p-values. Effect sizes provide an important metric for understanding the magnitude of differences beyond statistical significance, reinforcing the practical implications of our findings.

We believe this addition enhances the interpretability of our results and supports a more comprehensive discussion of the observed changes.

Comment 4:

Explicitly state how missing data (e.g., 2.28% in Table 7) was handled in analysis.

Reply:

We appreciate the reviewer's request for clarification regarding the handling of the missing data. To assess the randomness of missing values, we conducted Little’s MCAR test, which evaluates whether data are missing completely at random (MCAR) details are added under the heading of “Statistical Analysis”. The test results (χ² = 4.8987, p = 0.2979) confirmed that the missingness was random and did not introduce bias in our analysis. Given this confirmation, the missing data were handled appropriately without requiring further imputation or adjustments, ensuring the integrity of our statistical conclusions (also added in the statistical analysis section of the manuscript)

Comment 5:

Figures (1–4) are referenced but were not included in the PDF for evaluation (they are in separate TIFF files). Ensure that these are reviewed and captions are descriptive enough to be interpreted independently.

Reply:

Thank you for your observation. When uploading, we will ensure that all figures are included within the PDF file for evaluation. We have also reviewed the figures and provided descriptive captions to ensure they can be interpreted independently. To clarify, two figures have been added for Phase One and one figure for Phase Two (as per the comment of Reviewer # 1). Additionally, the previously labeled Figures 1 and 2 have been replaced with tables to improve clarity and presentation. As per PLOS ONE guidelines, figures are uploaded separately and are automatically included at the end of the compiled PDF file. We have also reviewed our previous submission, where the figures are appearing in the PDF after the main manuscript and before the supplementary material.

Comment 6:

The handoff tool (S1 File) should be briefly described (e.g., sections, items) in the text when first introduced.

Reply:

We have now included a brief description of the handoff tool (S1 File) in the text at its first mention.

Comment 7:

Minor polishing is needed for consistent tenses and clarity. For example: "Patients were predominantly transported by stretcher (48.13%) or walking (34.7%)" → "on foot" might be clearer. “...hospital-wide over the next two to three months” → specify actual timeline if possible.

Reply:

We have reviewed the manuscript to ensure consistent use of tenses and improved clarity throughout.

Comment 8:

Strengthen discussion with comparative insights from other QI projects in LMICs or outpatient-to-ED transition studies, if available.

Reply:

We found limited published data specifically addressing quality improvement initiatives focused on the transition of patients from outpatient clinics to emergency departments or other high acuity areas, particularly in low- and middle-income countries (LMICs).

To address this gap, we identified and included three pertinent studies that describe different models implemented within outpatient settings aimed at improving the management and transfer of critically ill patients to higher acuity care areas. These studies provide valuable frameworks for understanding effective strategies, challenges, and outcomes related to such transitions.

In the Discussion section, we have integrated these studies to provide comparative insights, highlighting similarities and differences with our intervention.

Comment 9:

Clarify whether the training program has ongoing refresher modules or sustainability plans post-study.

Reply:

We have added a new section titled “Sustainability Measures and Ongoing Staff Competency Reinforcement” after the phases. The training program is now part of the orientation for new nurses, and quarterly refresher sessions are held for existing staff to maintain key skills related to early warning signs, the handoff tool, and patient transfer protocols.

Comment 10:

Reply:

We have reviewed our financial disclosures and confirm that there are no changes. We have nothing to disclose and stated this in the cover letter in bold.

Comment 11:

If applicable, we recommend that you deposit your laboratory protocols in protocols.io to enhance the reproducibility of your results.

Reply:

Our study did not involve laboratory protocols requiring deposition in protocols.io.

Journal Requirement

Comment 1:

Reply:

We have reviewed and updated the manuscript and all associated files to ensure they meet PLOS ONE’s style and file naming requirements.

Comment 2:

We note that your Data Availability Statement is currently as follows: [All relevant data are within the manuscript and its Supporting Information files.]. Please confirm at this time whether your submission contains all raw data required to replicate the results of your study. Authors must share the “minimal data set” for their submission. PLOS defines the minimal data set to consist of the data required to replicate all study findings reported in the article, as well as related metadata and methods (https://journals.plos.org/plosone/s/data-availability#loc-minimal-data-set-definition).

Reply:

We are uploading the required data set with this revision under the file titled “Data Set”, in accordance with PLOS ONE guidelines. There will be two excel files: one for pre- and post-test data, and another for data on transferred patient outcomes

Comment 3:

Please review your reference list to ensure that it is complete and correct. If you have cited papers that have been retracted, please include the rationale for doing so in the manuscript text or remove these references and replace them with relevant current references. Any changes to the reference list should be mentioned in the rebuttal letter that accompanies your revised manuscript.

Reply:

We have reviewed the reference list and confirm that it is complete and accurate. None of the cited papers have been retracted. However, References 1 and 9, which were previously sourced from internet-based materials, have been replaced with citations from peer-reviewed journal articles for greater reliability.

Comment 4:

Reply:

Thank you for the guidance. We have uploaded the figure files to the PACE tool as instructed and have ensured they comply with PLOS formatting requirements.

Reviewer 1

Comment 1:

Title Improvement Needed: The title of the manuscript could be enhanced for clarity and relevance.

Reply:

We sincerely appreciate your valuable suggestion regarding the clarity and relevance of the manuscript title. In response, we have revised the title to better reflect the study’s objectives and scope. The original title,

"Rapid response and safety optimization: enhancing critical transfers from outpatient clinics to high acuity areas – a quality improvement project,"

has been replaced with:

"Ensuring Safe Transfer of Critically Ill Patients from Outpatient Clinics to High-Acuity Areas: A Quality Improvement Project."

We hope this revised title aligns better with the journal’s standards and expectations.

Comment 2:

Abstract

Method Clarity: The methodology is not clearly articulated in the abstract. A more precise definition of the phases and techniques used is necessary.

Reply:

In response to the request for greater clarity in the abstract methodology section, we have revised it to clearly define the study’s two phases, including the methods employed, timelines, and evaluation processes. This revision aims to adequately address the concern and improve overall clarity.

Comment 3:

Introduction

Problem Justification: The significance of the research problem should be elaborated upon. A clearer explanation of why this study is necessary would strengthen the introduction.

Reply:

We have expanded the introduction to provide a more detailed justification of the research problem, emphasizing the significance and necessity of this study. This enhancement aims to strengthen rationale and better contextualize the importance of our work.

Comment 4:

Methodology

General Explanation: A general overview of the phases should be presented in a dedicated figure within this section.

Reply:

We have added dedicated figures in the Methodology section that provides a clear overview of the study phases to enhance reader comprehension.

Comment 5:

Method Explanation: The overall methodology lacks clarity. A more thorough explanation is needed to understand the study's design and execution.

Reply:

We have revised the Methodology section to provide a more detailed and clear explanation of the study design and execution.

Comment 6:

Discussion

Findings-Based Discussion: The discussion section should initiate with a summary of the primary findings from the study.

Comparative Analysis: Include comparisons with countries having similar income levels or employing analogous methods, such as India and Iran. This context would enrich the discussion and provide a broader perspective.

Reply:

We began the Discussion section with a concise summary of the primary findings of our study. To the best of our knowledge and following a thorough literature search, we did not find any studies from low- and middle-income countries (LMICs) or neighboring countries that specifically address quality improvement initiatives focused on patient transitions from outpatient clinics to emergency or high-acuity areas relevant to our study title.

To address this gap, we identified and incorporated three relevant studies describing models used in outpatient settings to improve the management and transfer of critically ill patients to higher acuity care. These studies provide useful frameworks for understanding effective strategies, challenges, and outcomes in similar contexts.

These comparative insights have been integrated into the Discussion section to provide broader context and enrich the interpretation of our findings.

Comment 7:

Presentation

Schematic Representation: Processes should be illustrated in a schematic form to aid understanding.

Reply:

We have revised the manuscript according to your recommendation by adding a dedicated figure in the Methods section that provides a clear overview of the study phases and workflow.

Comment 8:

Conclusion

Ethical Declaration: A concluding declaration should be added to address ethics, budget considerations, author contributions, acknowledgments, and any necessary thanks.

Reply:

In response to your comment, the following have been addressed in the revised manuscript:

• A patient and public involvement declaration has been added after the Methods section as ethical declaration.

• Author contributions have been detailed separately in accordance with journal guidelines.

• An Acknowledgments section has been included in the revised manuscript.

---

## [Decision Letter · Decision Letter 1]

4 Aug 2025

Dear Dr. Fatima,

Thank you for submitting your manuscript to PLOS ONE. After careful consideration, we feel that it has merit but does not fully meet PLOS ONE’s publication criteria as it currently stands. Therefore, we invite you to submit a revised version of the manuscript that addresses the points raised during the review process.

We look forward to receiving your revised manuscript.

Kind regards,

Mohd Ismail Ibrahim, MCom.Med

Academic Editor

PLOS ONE

Journal Requirements:

Reviewers' comments:

Reviewer's Responses to Questions

**Comments to the Author**

Reviewer #1: (No Response)

Reviewer #2: (No Response)

Reviewer #3: All comments have been addressed

Reviewer #4: (No Response)

Reviewer #5: (No Response)

2. Is the manuscript technically sound, and do the data support the conclusions?

Reviewer #1: No

Reviewer #2: Yes

Reviewer #3: Yes

Reviewer #4: Yes

Reviewer #5: No

3. Has the statistical analysis been performed appropriately and rigorously?

Reviewer #1: No

Reviewer #2: Yes

Reviewer #3: Yes

Reviewer #4: Yes

Reviewer #5: No

4. Have the authors made all data underlying the findings in their manuscript fully available?

Reviewer #1: Yes

Reviewer #2: Yes

Reviewer #3: Yes

Reviewer #4: Yes

Reviewer #5: No

5. Is the manuscript presented in an intelligible fashion and written in standard English?

Reviewer #1: Yes

Reviewer #2: Yes

Reviewer #3: Yes

Reviewer #4: Yes

Reviewer #5: Yes

Reviewer #1: Dear authors, unfortunately, no changes have been made to the article, and the text has not been colored. This study has problems in the title, abstract, introduction, method, type of instrument, patient population, method of patient selection, and also in the selection of experts.

Reviewer #2: Review Comments to the Author

Thank you for the opportunity to review this well-conceived and practical quality improvement study, “Rapid response and safety optimisation: enhancing critical transfers from outpatient clinics to high acuity areas – a quality improvement project.” The manuscript addresses an important and underexplored aspect of patient safety in low-resource healthcare settings, specifically, the optimisation of intra-hospital transfers from outpatient clinics to high-acuity areas.

Overall, this is a technically sound, well-executed project that demonstrates a clear improvement in staff knowledge, structured communication, and aspects of patient transfer safety following the intervention. The manuscript is aligned with PLOS ONE’s scope, and the conclusions are appropriately drawn from the data presented. Below, I provide feedback to help enhance the manuscript further:

Strengths:

1. Timely and Relevant Topic: The focus on critical transfers from outpatient departments, particularly in a low- and middle-income country (LMIC), fills a noticeable gap in current literature and quality improvement discourse. The study is one of the few (and possibly the first) to focus specifically on improving intra-hospital transfers from outpatient departments in a tertiary LMIC setting.

2. Methodological Rigor: The quasi-experimental design, structured intervention (early warning criteria + handoff tool), and longitudinal data collection over a one-year period are appropriate and clearly described.

3. Operational Impact: The intervention is practical, feasible, and shows evidence of integration into routine clinical practice, which is a critical element of QI sustainability.

4. Ethical Considerations: The manuscript notes ERC exemption and informed consent, which is appropriate.

Recommendations for Improvement:

1. Contextualisation and Literature Framing

- Consider strengthening the Introduction to better highlight the gap in current research specific to outpatient-to-ED transfers. A brief contrast with inter-hospital or inpatient transfers in previous studies would underscore the manuscript’s novel contribution.

2. Clarity Around Causal Inference

- The current discussion frames the outcomes as improvements following the intervention, which is valid. However, given the lack of a control group, the language should be cautious in claiming causation. I suggest adding a brief note about this limitation in the Discussion section to strengthen the manuscript’s objectivity. Additionally, the Discussion occasionally restates earlier points without adding new insight.

3. Implementation Insights and Sustainability

- The manuscript mentions plans for digitisation and integration of handoff documentation. I recommend briefly describing any early steps toward this (if applicable) or a roadmap for future digitisation efforts to further highlight the potential for scale-up and sustainability.

4. Presentation and Writing Style

- The manuscript is generally clear and readable. I believe that minor grammatical corrections are needed, such as the consistency in terms like “handoff” vs. “handover”.

- Consider reducing redundancy in the Discussion and Conclusion sections, where some findings are restated without additional interpretation.

- I recommend referencing and providing descriptions for figures and tables, for instance, “Table 4 shows demographic breakdown…”.

5. Data Transparency

- The authors have adequately complied with PLOS ONE’s Data Availability Policy. The manuscript and supporting files contain sufficient data to validate and replicate key findings. Although raw individual-level data are not included, the aggregate and descriptive statistics are appropriately detailed.

6. Statistical Analysis

- The use of paired t-tests and descriptive analysis is appropriate. Assumptions were checked (Shapiro-Wilk test), and confidence intervals were reported. However, the interpretation of effect size or clinical relevance for some metrics, like the change in HR/RR, could be discussed further in the Results or Discussion.

Conclusion

This is a valuable contribution to the literature on patient safety and health systems strengthening in LMICs. The authors have implemented a feasible and impactful intervention. It was a good read.

Reviewer #3: Minor claculation mistake in line number 301 percentage of female nursing staff should be 77.5 instead of 80. and in line number 318: table 4. row number 3 and 4, 2nd column Female % should be 77.5 and Male 22.5.

Therefore, need this to be corrected. Rest seems OK.

Reviewer #4: This manuscript was written in an intelligible manner, using the acceptable standard of English. However, minor corrections such as punctuation marks and grammar should be modified thoroughly. Regarding the publication ethics, even though this is a QI study, PLOS ONE requires ethics info. You can write:

“This quality improvement study was approved as a non-research activity by the Ethical Review Committee of Aga Khan University Hospital, which waived the need for informed consent as no patient-identifiable data was used.”

It will also be important to mention the study code and the date on which the ethical clearance was issued.

In case you did not receive the ethical clearance, you must explain why you did not obtain formal ethics approval and under what guidelines it was exempted.

No proof of dual publication was noted.

Reviewer #5: Good day

Thank you for asking me to review the clinical paper

Overall, the manuscript presented a very important study aiming at improving the care and safety of patients in acute care episode. However, the manuscript has not been well arranged and the chosen study design was not properly followed

1. It was not clear whether the design is quazi experimental study, descriptive cross sectional or retrospective medical record review. The manuscript has characteristics depicting all the three

2. It is not clear whether this is a pilot study or a full scale study conducted after the preliminary pilot study

3. Pre and post quazi experimental design was mixed up, the intervention period is not clear. Especially, phase II consist of both intervention and outcome evaluation

4. many tools were mentioned and the effectiveness of the tools for early detection of patient deterioration were not discussed

5. Post intervention evaluation was not clear as there was no comparison between the pre and post. Therefore, results were ambiqous

6. The tables are many and mechanical. It does not show evidence of quazi experimental study conducted. It was mere listing of materials/instruments

7. Study setting, population, sample size and procedure for data collection not clear

8. How confounding variables were controlled not discussed

9. There was no clear evidence of testing a particular tool effectiveness.

10. Vital signs are known parameters to determine the degree of wellness or illness of patients. It is not appropriate to test vital sign usefulness or effectiveness. We know already

Summary

I. Re visit the study design

2. Discuss the baseline assessment only in phase I and the assessment tool to test their/its efficacy

3. Discuss the Interventions

4. Discuss the outcome evaluation to determine the effectiveness of the applied tools

Compare your results pre and post and make recommendations

5. Most of the tables presented are not necessary. Present the pre assessment (Phase 1)tables, the Interventions tables, post intervention table. Also present tables on association or correlation between the post and interventions etc

6. Describe each of the research step clearly: The population, sampling, sample size, elligibility criteria, validity, reliability etc

7. You may combine few related results/tables

8. Make your discuss coincide, identify what this study add or concluded, compare with previous studies, state the strength and limitations of the study

**Do you want your identity to be public for this peer review?** For information about this choice, including consent withdrawal, please see our Privacy Policy

Reviewer #1: No

Reviewer #2: **Yes: ** Ella C. Nelson, MD, MBA, FISqua

Reviewer #3: No

Reviewer #4: **Yes: ** Siyakudumisa Nontamo

Reviewer #5: **Yes: ** Dr Ogundeji KD

---

## [Author Response · Author response to Decision Letter 2]

28 Sep 2025

Subject: Revised Submission – Manuscript ID PONE-D-25-12396R1

Dear Dr. Mohd Ismail Ibrahim,

Thank you for your detailed feedback and for giving us more time to revise our manuscript. We are grateful to you and the reviewers for your constructive comments, which have helped us improve the clarity, rigor, and presentation of our work.

In response to the reviewers’ suggestions, we have:

• Addressed each comment in detail in the attached “Response to Reviewers” document.

• Revised the manuscript to incorporate all necessary changes, including methodological clarifications, improved discussion points, and updated references where appropriate.

• Provided both a marked-up version showing tracked changes and a clean version of the revised manuscript.

We believe the revisions have strengthened the manuscript and addressed all concerns raised during the review process. We appreciate your consideration of our revised submission and look forward to your feedback.

Kind regards,

Dr. Samar Fatima

Journal Requirements

Comment 1:

Reply 1:

We would like to confirm that there have been no changes to the financial statements previously provided, as referenced in the cover letter.

All figure files have been checked to ensure they meet PLOS ONE submission requirements.

Comment 2:

If applicable, we recommend that you deposit your laboratory protocols in protocols.io to enhance the reproducibility of your results. Protocols.io assigns your protocol its own identifier (DOI) so that it can be cited independently in the future. For instructions, see: https://journals.plos.org/plosone/s/submission-guidelines#loc-laboratory-protocols. Additionally, PLOS ONE offers an option for publishing peer-reviewed Lab Protocol articles, which describe protocols hosted on protocols.io. Read more information on sharing protocols at https://plos.org/protocols?utm_medium=editorial-email&utm_source=authorletters&utm_campaign=protocols.

Reply 2:

Our study does not involve laboratory protocols requiring deposition in protocols.io. Therefore, this recommendation does not apply to the present manuscript.

Reviewer #1

Comment:

Dear authors, unfortunately, no changes have been made to the article, and the text has not been colored. This study has problems in the title, abstract, introduction, method, type of instrument, patient population, method of patient selection, and also in the selection of experts.

Reply:

In view of Reviewer 1’s above comment, we believe that there may have been a misunderstanding and that the reviewer may have inadvertently reviewed an older version of our manuscript rather than the revised submission with tracked changes.

We appreciate the Academic Editor’s review of our concerns and confirmation that substantial changes were made in response to Reviewer 1’s original feedback. We also acknowledge the Editor’s assessment that the comment “no changes have been made” does not align with the actual revisions incorporated, and that this may have resulted from difficulty accessing the track-changed document.

The link to the Academic Editor’s full email response confirming this matter has also been provided here for reference: [Academic Editor Response].

We appreciate the Editor’s effort in seeking additional reviewer input to ensure a balanced and fair evaluation. Accordingly, we will proceed to address the remaining reviewer and editor comments to further improve the manuscript.

Reviewer #2:

Comment:

Review Comments to the Author

Thank you for the opportunity to review this well-conceived and practical quality improvement study, “Rapid response and safety optimization: enhancing critical transfers from outpatient clinics to high acuity areas – a quality improvement project.” The manuscript addresses an important and underexplored aspect of patient safety in low-resource healthcare settings, specifically, the optimization of intra-hospital transfers from outpatient clinics to high-acuity areas.

Overall, this is a technically sound, well-executed project that demonstrates a clear improvement in staff knowledge, structured communication, and aspects of patient transfer safety following the intervention. The manuscript is aligned with PLOS ONE’s scope, and the conclusions are appropriately drawn from the data presented. Below, I provide feedback to help enhance the manuscript further:

Strengths:

1. Timely and Relevant Topic: The focus on critical transfers from outpatient departments, particularly in a low- and middle-income country (LMIC), fills a noticeable gap in current literature and quality improvement discourse. The study is one of the few (and possibly the first) to focus specifically on improving intra-hospital transfers from outpatient departments in a tertiary LMIC setting.

2. Methodological Rigor: The quasi-experimental design, structured intervention (early warning criteria + handoff tool), and longitudinal data collection over a one-year period are appropriate and clearly described.

3. Operational Impact: The intervention is practical, feasible, and shows evidence of integration into routine clinical practice, which is a critical element of QI sustainability.

4. Ethical Considerations: The manuscript notes ERC exemption and informed consent, which is appropriate.

Recommendations for Improvement:

1. Contextualisation and Literature Framing

- Consider strengthening the Introduction to better highlight the gap in current research specific to outpatient-to-ED transfers. A brief contrast with inter-hospital or inpatient transfers in previous studies would underscore the manuscript’s novel contribution.

2. Clarity Around Causal Inference

- The current discussion frames the outcomes as improvements following the intervention, which is valid. However, given the lack of a control group, the language should be cautious in claiming causation. I suggest adding a brief note about this limitation in the Discussion section to strengthen the manuscript’s objectivity. Additionally, the Discussion occasionally restates earlier points without adding new insight.

3. Implementation Insights and Sustainability

- The manuscript mentions plans for digitisation and integration of handoff documentation. I recommend briefly describing any early steps toward this (if applicable) or a roadmap for future digitisation efforts to further highlight the potential for scale-up and sustainability.

4. Presentation and Writing Style

- The manuscript is generally clear and readable. I believe that minor grammatical corrections are needed, such as the consistency in terms like “handoff” vs. “handover”.

- Consider reducing redundancy in the Discussion and Conclusion sections, where some findings are restated without additional interpretation.

- I recommend referencing and providing descriptions for figures and tables, for instance, “Table 4 shows demographic breakdown…”.

5. Data Transparency

- The authors have adequately complied with PLOS ONE’s Data Availability Policy. The manuscript and supporting files contain sufficient data to validate and replicate key findings. Although raw individual-level data are not included, the aggregate and descriptive statistics are appropriately detailed.

6. Statistical Analysis

- The use of paired t-tests and descriptive analysis is appropriate. Assumptions were checked (Shapiro-Wilk test), and confidence intervals were reported. However, the interpretation of effect size or clinical relevance for some metrics, like the change in HR/RR, could be discussed further in the Results or Discussion.

Conclusion

This is a valuable contribution to the literature on patient safety and health systems strengthening in LMICs. The authors have implemented a feasible and impactful intervention. It was a good read.

Reply:

We sincerely thank the reviewers for their thoughtful, encouraging, and constructive feedback, and for recognizing the value and relevance of our work. We greatly appreciate the time and effort taken to provide detailed recommendations for improvement. We address each point below and have incorporated the suggested changes into the revised manuscript (with tracked changes provided).

1. Contextualization and Literature Framing

In the Introduction, we have added paragraphs 2 and 3 to incorporate relevant literature on inter- and intra-hospital transfers. Despite an extensive search, we were unable to identify any published studies specifically examining transfers from outpatient or clinic settings to emergency departments in low- and middle-income countries (LMICs) or South Asia, including Pakistan. Existing literature from the region largely focuses on inter-facility (hospital-to-hospital) transfers, ambulance and prehospital services, and the epidemiology of emergency department presentations, but does not address outpatient-to-ED transfer processes or outcomes.

Please note that we are referring to paragraph and line numbers rather than absolute line numbers, as the latter may differ between the clean and tracked-changes versions.

2. Clarity Around Causal Inference

We are making a clear statement in the limitation section of the discussion (4th line). We have revised the Discussion to minimize repetition and ensure that each point provides new insight or interpretation of the findings.

3. Implementation Insights and Sustainability

We have added a brief description of early steps and plans for digitization and integration of handoff documentation in the Discussion (7th paragraph, line 8) to highlight the potential for scale-up and sustainability.

4. Presentation and Writing Style

We have made every effort to reduce grammatical errors. As mentioned above, we have revised the entire Discussion/Conclusion and added additional interpretations. Furthermore, we have referenced and provided descriptions for all figures and tables while ensuring compliance with PLOS ONE requirements.

5. Data Transparency

We have provided complete data, which is included in the file inventory, including all pre- and post-test data as well as the patient dataset. Should you require any additional specific information, we would be happy to provide it.

6. Statistical Analysis

We appreciate the reviewer’s comment. The significance of effect sizes has been added to the Results section under the pre-test and post-test headings (Paragraph 2).

Additionally, the interpretation of changes in HR and RR has been added under 'Outcomes of patient transfers following the implementation of the quality improvement initiative' in the results section (Paragraph 2).

Reviewer #3

Comment:

Minor calculation mistake in line number 301, the percentage of female nursing staff should be 77.5 instead of 80. and in line number 318: table 4. row number 3 and 4, 2nd column Female % should be 77.5 and Male 22.5.

Therefore, need this to be corrected. Rest seems OK.

Reply:

Thank you very much for your careful review and kind guidance. We sincerely appreciate you pointing out the calculation errors. I have now corrected the percentages as recommended.

Reviewer #4

Comment:

This manuscript was written in an intelligible manner, using the acceptable standard of English. However, minor corrections such as punctuation marks and grammar should be modified thoroughly.

Regarding publication ethics, even though this is a QI study, PLOS ONE requires ethics info. You can write:

“This quality improvement study was approved as a non-research activity by the Ethical Review Committee of Aga Khan University Hospital, which waived the need for informed consent as no patient-identifiable data was used.”

It will also be important to mention the study code and the date on which the ethical clearance was issued.

In case you did not receive the ethical clearance, you must explain why you did not obtain formal ethics approval and under what guidelines it was exempted.

No proof of dual publication was noted.

Reply:

We thank the reviewer for the suggestion and have corrected grammatical and punctuation errors throughout the manuscript.

Regarding publication ethics, we would like to clarify that, in accordance with institutional policy at Aga Khan University Hospital, all research and quality improvement projects undergo Ethical Review Committee (ERC) oversight. For this project, permission was first obtained from the Chief Medical Officer as well, and the ERC subsequently granted an exemption. No patient-identifiable data was used, and written informed consent was obtained from participating nurses and healthcare staff for pre- and post-test assessments. The ERC exemption letter is included in the inventory files.

While we appreciate the reviewer’s suggested wording, we have retained the phrasing above to accurately reflect our institutional process and policies. We have also added a statement confirming that this manuscript is original, has not been published elsewhere, and no dual publication has occurred.

Reviewer #4: Reviewer 4 additional comments from the Word document sent as an attachment:

General Comments

This is a well-organized and useful Quality Improvement (QI) project that aimed to improve the safety and effectiveness of patient flow from outpatient clinics to the ED at a tertiary care setting in Pakistan. The subject is especially pertinent in low- and middle-income countries (LMICs), where healthcare systems operate under varying challenges specific to each setting. The authors discuss a topic that is often overlooked when considering patient safety, and the interventions described are relatively pragmatic and potentially scalable. However, there are several areas where improvements would enhance the manuscript's scientific rigor, clarity, and impact.

Specific Comments

1. Title & Abstract

The title is well structured, but could be concise, such as

“Improving the Safety of Outpatient to Emergency Department Transfers: A Quality Improvement Study in a Tertiary Hospital in Pakistan”

The abstract follows a logical structure, yet needs keywords like quality improvement, handover, and patient safety to improve search results

2. Introduction

The similar initiatives as part of the literature supporting the relevance of your study were not cited sufficiently, such as QI initiatives in similar LMIC settings (abroad and local).

- Please look for studies in QI Initiatives conducted in similar settings to support your argument.

Reword or rephrase for conciseness and impact, maybe to write:

‘Transfers from outpatient clinics to the Emergency Department are vulnerable points in the care continuum, especially in low and middle-income countries, where the standardized protocols are often lacking’

In this introduction, the problem and its significance were not clearly defined.

- Please define clearly the problem and its significance.

- Secondly, support your problem by mentioning the available data on occurrences, like noting that:

‘? % of patients reported missing paperwork…’

It is always recommended that your introduction ends with the purpose or aim of conducting the study, e.g.:

"This study aimed to improve the safety, documentation, and tracking of patients transferred from outpatient clinics to the ED through a quality improvement approach at Aga Khan University Hospital.

3. Methods

3.1 Study Design and Setting

This is not a clinical trial. Therefore, mention that this is a QI initiative to minimize confusion for the readers.

When conducting a Quality Improvement study in healthcare, the SQUIRE 2.0 (Standards for Quality Improvement Reporting Excellence) is often used for reporting.

SQUIRE 2.0 provides a framework for authors to report their QI projects clearly and transparently.

Therefore, I recommend that you use the SQUIRE 2.0 structure to address the following 3 areas, namely;

a) Context

Describe the hospital setting (for instance, yearly patient numbers, number of specialties)

b) Intervention rationale:

Explain why these specific tools, such as the transfer form and clinic attendant, were chosen.

c) Measures:

Lastly,

---

## [Decision Letter · Decision Letter 2]

24 Oct 2025

Improving the safety of outpatient to Emergency Department transfers: A quality improvement study in a tertiary hospital in Pakistan

PONE-D-25-12396R2

Dear Dr. Fatima,

We’re pleased to inform you that your manuscript has been judged scientifically suitable for publication and will be formally accepted for publication once it meets all outstanding technical requirements.

Kind regards,

Mohd Ismail Ibrahim, MCom.Med

Academic Editor

PLOS ONE

Additional Editor Comments (optional):

Please note that this manuscript was reviewed by three peer reviewers. However, only two of them were deemed to have provided valid and constructive feedback for improving the manuscript. I have carefully examined all the reviewers’ comments and am satisfied with the revisions made. Therefore, I conclude that the manuscript is suitable for acceptance and publication.

Reviewers' comments:

Reviewer's Responses to Questions

**Comments to the Author**

Reviewer #1: (No Response)

Reviewer #2: All comments have been addressed

Reviewer #4: All comments have been addressed

2. Is the manuscript technically sound, and do the data support the conclusions?

Reviewer #1: No

Reviewer #2: Yes

Reviewer #4: Yes

3. Has the statistical analysis been performed appropriately and rigorously?

Reviewer #1: No

Reviewer #2: Yes

Reviewer #4: Yes

4. Have the authors made all data underlying the findings in their manuscript fully available?

Reviewer #1: Yes

Reviewer #2: Yes

Reviewer #4: Yes

5. Is the manuscript presented in an intelligible fashion and written in standard English?

Reviewer #1: Yes

Reviewer #2: Yes

Reviewer #4: Yes

Reviewer #1: Dear authors,

the following suggestion is provided.

Abstract:

In the method, while specifying the type of study, the method should be stated more precisely. The dates should be stated as time efficiency.

In the introduction,

Objectives should not be categorized and should be stated in the introduction.

Context should be stated more briefly.

Quality improvement initiative. Explain this with reference.

The method is not categorized well.

The description in the method is too long. It should be categorized in a precise study category.

The discussion should be examined in more depth with the objectives of the study.

Ethical considerations in the research have not yet been stated.

Regarding the budget of the subject proposal, it has not been stated.

Reviewer #2: I am pleased that the authors have responded thoughtfully to prior feedback, improving the structure, clarity, and methodological transparency of the protocol. Overall, the revisions address the major concerns from the previous round. The study is well-conceived, and the improvements demonstrate careful attention to reviewer feedback.

Reviewer #4: All the concerns and recommendations from the reviewer were adequately addressed. In addition, one of the major concerns of the reviewer was the omission of the ethical approval (study code, date of issue, and the issuing body). This has been sufficiently provided. The modification of the topic/title was also attended to. Similarly, the clear distinction that this study is QI INITIATIVE and not CLINICAL TRIAL improved the conciseness and clarity. Lastly, the study was written in an intelligible fashion using standard English.

**Do you want your identity to be public for this peer review?** For information about this choice, including consent withdrawal, please see our Privacy Policy

Reviewer #1: No

Reviewer #2: **Yes: ** Ella C. Nelson | MD | MBA | FISQua

Reviewer #4: **Yes: ** Siyakudumisa Nontamo

---

## [Editor Report · Acceptance letter]

PONE-D-25-12396R2

PLOS ONE

Dear Dr. Fatima,

I'm pleased to inform you that your manuscript has been deemed suitable for publication in PLOS ONE. Congratulations! Your manuscript is now being handed over to our production team.

Kind regards,

on behalf of

Dr. Mohd Ismail Ibrahim

Academic Editor

PLOS ONE